# Controlled formation of three-dimensional cavities during lateral epitaxial growth

Yiwen Zhang [1,4], Baoming Wang [2,4], Changxu Miao [3,4], Haozhi Chai [1], Wei Hong [3] ✉, Frances M. Ross [2] ✉ & Rui-Tao Wen [1] ✉

Epitaxial growth is a fundamental step required to create devices for the semiconductor industry, enabling different materials to be combined in layers with precise control of strain and defect structure. Patterning the growth substrate with a mask before performing epitaxial growth offers additional degrees of freedom to engineer the structure and hence function of the semiconductor device. Here, we demonstrate that conditions exist where such epitaxial lateral overgrowth can produce complex, three-dimensional structures that incorporate cavities of deterministic size. We grow germanium on silicon substrates patterned with a dielectric mask and show that fully-enclosed cavities can be created through an unexpected self-assembly process that is controlled by surface diffusion and surface energy minimization. The result is confined cavities enclosed by single crystalline Ge, with size and position tunable through the initial mask pattern. We present a model to account for the observed cavity symmetry, pinch-off and subsequent evolution, reflecting the dominant role of surface energy. Since dielectric mask patterning and epitaxial growth are compatible with conventional device processing steps, we suggest that this mechanism provides a strategy for developing electronic and photonic functionalities.

Homo- and heteroepitaxial growth are fundamental processing steps required to develop semiconductor and quantum technologies. After the growth of the epitaxial stack, post-processing via top-down fabrication process steps, such as patterning and etching, are normally employed to construct devices with desired properties. In contrast, bottom-up approaches take advantage of physicochemical processes that arise spontaneously. One example, selective area epitaxy (SAE), in which preferential deposition and epitaxy take place on regions of the substrate defined by a mask layer, has become a versatile route to achieve various building blocks from single crystalline epilayers[1–6]. Once the epilayer has grown thicker than the mask (made most commonly of the dielectrics $SiO_2$ or $Si_3N_4$), further growth continues as epitaxial lateral overgrowth (ELO). SAE and ELO have been studied in

detail in group IV[7–12] and group III-V[13–16] systems because they provide pathways to improve the crystal quality of the epilayer. A common strategy involves the use of a thick mask layer with openings in the shape of long parallel strips. As the epilayer grows through the mask openings, threading dislocation densities are reduced due to a dislocation trapping effect; as growth extends laterally, growth fronts merge over the mask[7–12,17,18]. This process can result in a high crystal quality epilayer although elongated, hemicylindrical voids[7–12] may remain at locations where the growth fronts meet. This phenomenon is attributed to the shape of the growth front at coalescence[7–10] and may be useful in trapping dislocations[9].

The success of these SAE and ELO strategies has spurred substantial effort in developing optoelectronic and quantum devices. For

[1]Department of Materials Science and Engineering, Southern University of Science and Technology, 518055 Shenzhen, Guangdong, China. [2]Department of Materials Science and Engineering, Massachusetts Institute of Technology, Cambridge, MA 02139, USA. [3]Shenzhen Key Laboratory of Soft Mechanics and Smart Manufacturing, Department of Mechanics and Aerospace Engineering, Southern University of Science and Technology, 518055 Shenzhen, Guangdong, China. [4]These authors contributed equally: Yiwen Zhang, Baoming Wang, Changxu Miao. ✉e-mail: hongw@sustech.edu.cn; fmross@mit.edu; wenrt@sustech.edu.cn

example, InGaAs nanowire arrays grown by SAE on Si substrates are the basis for high-performance vertical transistors[19]. Millimeter-sized arrays of doped InP nanowires on patterned substrates demonstrate record solar efficiency of 13.8%, ref. [20]. Based on SAE and lateral epitaxy, an InAs/Al semiconductor/superconductor structure has been obtained[21]. Lateral epitaxy can even be used to modify crystal structure: Ge and SiGe alloys can be grown with hexagonal stacking and hence with a direct bandgap emission[22] through lateral epitaxy on a wurtzite-GaAs core, offering opportunities for Si- and Ge-based photonics and light emission[23].

These growth studies and device designs[6–12,17,19,20] involved continuous masks with elongated parallel openings (Supplementary Fig. 1). ELO on isolated mask areas is less studied[24]. This is partially because the structures and associated benefits, such as dislocation reduction and strain relief, are most straightforward to realize for elongated parallel mask openings[19,20]. For isolated mask areas, lateral growth of the epilayer beyond the mask edge leads to impingement of growth fronts from different directions, potentially creating more complex structures than are grown using continuous masks with parallel openings[9,13,24–29].

Here, we demonstrate that by taking advantage of the intrinsic dynamics of ELO, confined cavities in single crystalline Ge can be created with controllable size and position, bounded by internal facets aligned to Si crystallographic directions, through the use of an isolated mask. We measure the growth progression to show that the curvature of the mask edge leads to accelerated ELO: thus, although Ge ELO starts anisotropically due to the varied shape of the mask, it gradually transitions to an isotropic growth with a faceted but approximately circular boundary. Most strikingly, continued growth causes this growth front to coalesce, closing up above the isolated mask area to create an enclosed, faceted cavity without dislocations at the coalescence point. The shape of the cavity in three dimensions is consistent for varied mask shapes, and different sizes are achievable. We develop a phase-field model to quantify the progression of the growth front, replicating the anisotropic-isotropic transition and the pinch-off process that forms the cavities, which we show is dominated by surface energy minimization. The resulting process, with geometry that is distinct from that arising from elongated parallel mask openings, provides an additional degree of freedom to engineer device

structures through a bottom-up approach, relevant to applications such as light trapping in photonics.

## Results

### Anisotropic-isotropic transition and cavity formation

Si (001) wafers were first covered with thermal oxide of ~40 nm thickness to act as the mask material. Isolated areas with various sizes and shapes were defined through lithography, as described in "Methods". Subsequently, Ge epitaxy was achieved using chemical vapor deposition, with Ge growing only on the exposed Si area. To avoid Stranski-Krastanov growth, in which Ge grows as islands instead of the desired planar layer, a two-step growth recipe was used[24,30]. Growth at high temperature yields both a high deposition rate and high mobility of Ge adatoms. Lower temperature growth (see Supplementary Fig. 2) does not produce favorable morphologies because of the low mobility of Ge adatoms. Instead, the growth front is irregular with an acute angle (an obtuse angle is a prerequisite to form a cavity) and overgrowth to an anisotropic-isotropic transition was not visible. Various mask shapes were defined with dimensions in the range of 0.4–7.5 μm (Supplementary Fig. 3) and deposition was terminated at different stages to extract the ELO dynamics through scanning and transmission electron microscopy (SEM and TEM). Figure 1a and b show a 3D schematic illustration and associated top-view SEM images of the ELO process over round masks and squares with edges aligned to Si [110]. The images reveal that, for a round mask, the ELO initiates at all edges simultaneously. Thus, ELO is overall isotropic at the beginning. This is maintained until coalescence, as illustrated in the upper panels of Fig. 1a and b, thus the speed of lateral growth is equal in all directions. For a square mask, ELO is fastest initially at the four corners where curvature is maximized. Although it is anisotropic at the beginning, growth gradually transitions to isotropic as ELO proceeds, as seen in the circular shape of the Ge growth front over the square mask (lower panels of Fig. 1a and b). We find similar anisotropic-isotropic transitions in other symmetrical masks (i.e., triangular, quadrilateral, pentagonal, hexagonal and cross-shaped masks, Supplementary Fig. 4), and this transition occurs irrespective of the aligned angle between mask and Si [110] (Supplementary Fig. 5). The characteristic time of the anisotropic-isotropic transition varies with the size and shape (Supplementary Fig. 5). Theoretical models, developed below to

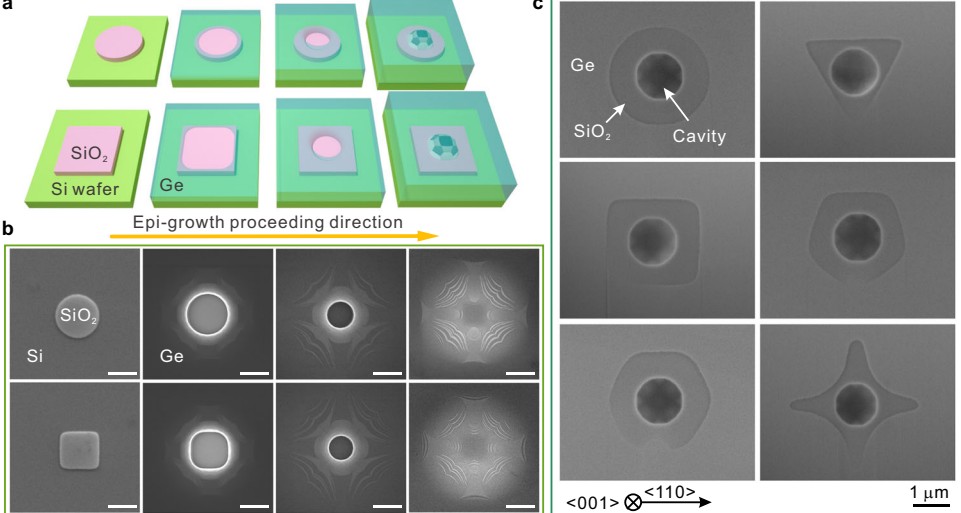

**Fig. 1 | Epitaxial lateral overgrowth (ELO) of Ge on SiO₂-patterned Si (001) substrate. a** Schematic illustration of the Ge epi-growth process. A faceted cavity forms below the coalescence position when the growth fronts eventually converge. **b** Top-view SEM images of Ge growth after epitaxy at the stages corresponding to a. The scale bar is 2.0 μm. The striped contrast on the Ge surface suggests a faceted top surface, as shown in Fig. S5c. The thickness of SiO₂ is ~40 nm and the square SiO₂ mask edges are aligned along Si [110]. **c** SEM images of Ge/SiO₂, viewed from beneath after removal of the Si substrate as described in Fig. S7, revealing a similar cavity at the center of each shape of the SiO₂ mask, and cavity edges parallel to Si [110].

understand the underlying mechanisms, have successfully reproduced these observations. As the isotropic regime of ELO proceeds, the growth front over the mask shrinks and finally pinches off, leaving a cavity beneath the coalescence point, as seen in the schematic polyhedron in the right part of Fig. 1a and the plan-view SEM images for various mask patterns in Fig. 1c. The experimental procedure for obtaining this cavity shape is shown in Supplementary Figs. 6 and 7. Independent of the mask geometry or the alignment angle between mask and Si [110], cavities with a consistent dodecagonal base form in the center of the masks, as shown in Fig. 1c and Supplementary Fig. 8. To verify these results, we checked at least 3 samples for each mask shape and size, which yielded consistent diameters and profiles for the formed cavities. An important feature of the growth process is that the ELO Ge above the cavity is found to be dislocation-free at the coalescence point, based on statistical information from over 10 TEM samples with different mask shapes (Supplementary Fig. 9). Between mask regions, threading dislocations are observed at about $2 \times 10^7 \, \text{cm}^{-2}$ (a value consistent with our previous measurements from vertical epi-Ge on Si[31]) that are associated with misfit dislocations originating at the Ge/Si interface (Supplementary Fig. 9). The ELO Ge over the mask therefore offers, as a byproduct of the lateral growth and coalescence, an opportunity to achieve dislocation-free Ge, at least in microscale regions.

## Phase field modeling of anisotropic to isotropic transition and cavity pinch-off

We develop a phase-field model to reveal the underlying mechanisms of the anisotropic-isotropic transition and the meeting of the growth fronts to form the cavity. Some representative results are shown in Fig. 2, and the model is described further in Supplementary Fig. 10. Ge atoms in the system may undergo four kinetic processes: (1) diffusion in the gas phase, (2) attachment on the solid surface, (3) migration on the gas-solid interface as an adatom, and (4) diffusion in the solid phase. A continuous phase field $C(\mathbf{x}, t)$ is introduced to distinguish the gas phase ($C = 0$) and the solid phase ($C = 1$), and to trace the morphological and topological evolution, as shown in Supplementary Fig. 11. In contrast to a sharp interface model, a phase field model does not require explicitly tracking the motion of interfaces, and thus facilitates modeling of complex morphological and even topological evolution. Following common practice in phase-field modeling, we introduce a double-well function $\Phi_{DW} = \eta C^2 (C - 1)^2$ to enable phase separation, and a gradient term $\Phi_G = \frac{\gamma}{2} |\nabla C|^2$ to mediate the abrupt but continuous transition between the two phases, and we write the bulk free energy density in the form

$$\Phi = \Phi_{DW} + \Phi_G = \eta C^2 (C - 1)^2 + \frac{\gamma}{2} \frac{\partial C}{\partial x_i} \frac{\partial C}{\partial x_i}, \quad (1)$$

with repeated indices indicating a summation.

A comparison between the coefficients of the two contributions yields a length $L_0 = \sqrt{\gamma / 2\eta}$, which characterizes the thickness of the transition layer between the gas and solid phases. Here, an isotropic and orientation-independent surface energy is adopted to reduce computational cost in most numerical simulations, except for the simulations showing a faceted surface. These are obtained through an extended model in which the constant surface energy $\gamma$ is replaced by

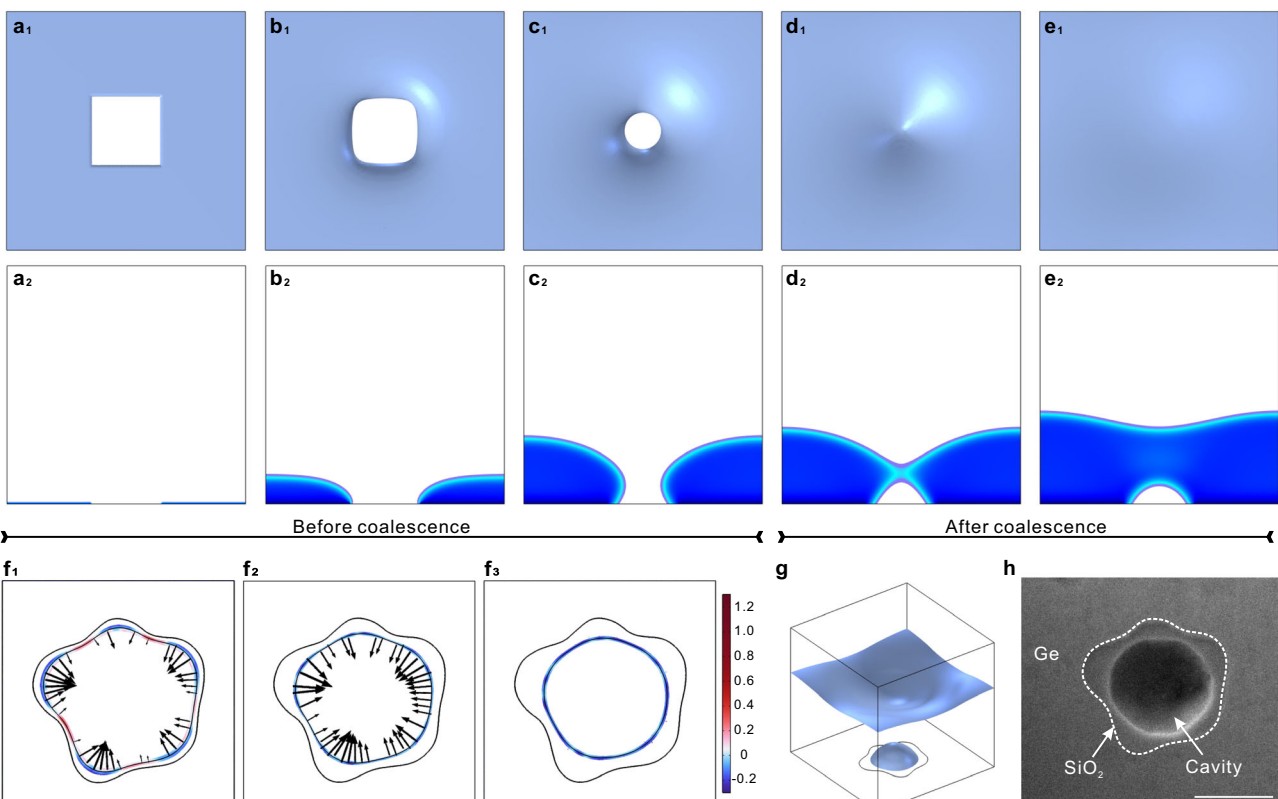

**Fig. 2 | Simulation of anisotropic-isotropic evolution and cavity formation on masks.** $a_1$–$e_1$ Top-view snapshots of the profile evolution of the Ge surface during epi-growth over a square mask. The shape of the opening transitions from the mask shape to a circle while gradually reducing in size until closing. $a_2$–$e_2$, Side-view snapshots corresponding to $a_1$–$e_1$. A dome-like cavity is formed at the center of the mask. $f_1$–$f_3$ Surface-tension-dominated growth of the irregular front over a random geometry mask. The arrows indicate the interface advancing velocity, and the color represents the mean curvature. **g** 3D view of the surface profile after the ELO layer pinches off over an asymmetric mask. The solid blue, representing solid Ge, is removed to reveal the shape of the oxide. **h** SEM image of the Ge/mask interface, revealing the cavity over the center of the asymmetric mask. The scale bar is 1.0 μm.

an orientation-dependent term, as detailed in Supplementary Note 4. Numerical tests have not shown significant changes in the modeled phenomena, except for the faceted surface at equilibrium.

To account for mass conservation in the epi-growth process, we directly associate the phase field $C$ with the concentration of Ge atoms, and write the evolution equation in the form

$$\frac{\partial C}{\partial t} = -\frac{\partial j_i}{\partial x_i}, \tag{2}$$

with flux vector $j_i$ related to the gradient of chemical potential $\mu$ as

$$j_i = -M_{ik}\frac{\partial \mu}{\partial x_k}, \tag{3}$$

**M** being the mobility tensor, and repeated indices indicating summations. The chemical potential, on the other hand, is calculated from the variation of the bulk free energy density with respect to $C$:

$$\mu = \frac{\delta \Phi}{\delta C} = 2\eta C\left(2C^2 - 3C + 1\right) - \gamma\frac{\partial^2 C}{\partial x_i \partial x_i}. \tag{4}$$

It can be seen that the flux, driven by both the concentration gradient of Ge atoms and the surface-energy-induced Laplace pressure, is proportional to the mean curvature of the surface as given by the second term on the right-hand side of Eq. (4), refs. 32,33. Here, for consistency with that in the solid phase, the transportation of Ge atoms in the gas phase is also modeled as a diffusion process. As the focus of the current model is the morphological evolution of the solid phase, the actual transportation process of Ge in the gas phase is deemed to be a minor factor, as long as the deposition flux is properly prescribed and the topology of the gas domain accounted for.

Unlike sharp-interface models, phase-field models consider interface and bulk regions as a continuous system, rather than treating them as separate entities. Therefore, conventional phase-field models do not distinguish between surface and bulk diffusion. Here, to specially account for the contribution from surface diffusion, we write the mobility tensor into an anisotropic form

$$M_{ik} = M_\mathrm{B}\delta_{ik} + M_\mathrm{S}\left(\frac{\partial C}{\partial x_j}\frac{\partial C}{\partial x_j}\delta_{ik} - \frac{\partial C}{\partial x_i}\frac{\partial C}{\partial x_k}\right). \tag{5}$$

The first term on the right-hand side of Eq. (5) represents the mobility of bulk diffusion, $M_\mathrm{B}$, interpolated smoothly between a higher value in the gas phase and a much lower level in the solid phase. The second term captures the surface diffusion within the transition zone between the gas and solid phases, where $|\nabla C|$ takes non-vanishing values. A transversely isotropic tensor with vanishing value along the interface normal is assumed to limit the contribution to the fluxes on the interface (more details in Supplementary Note 1).

The different wettability of Ge on Si and SiO$_2$ surfaces is accounted for through boundary conditions. On the non-wetting SiO$_2$ surface, the normal gradient of the phase field is prescribed by

$$\frac{\partial C}{\partial x_i}n_i = \sqrt{\frac{2\Phi_\mathrm{DW}}{\gamma}}\cos\alpha = \frac{1}{L_0}C(1-C)\cos\alpha, \tag{6}$$

with $\alpha \approx 120°$ being the static contact angle of Ge on SiO$_2$, ref. 34 (see Supplementary Fig. 12 and Supplementary Note 2). Such a boundary condition does not prevent Ge adatoms landing on the SiO$_2$ surface, but effectively increases the free energy by enforcing a solid-gas interface with the prescribed contact angle. On the other hand, $C = 1$ is prescribed on the wetting Si surface. With impermeable conditions on all sidewalls, a constant diffusion flux is prescribed atop the entire computational domain to simulate the ELO process.

We start with the simulation of a representative ELO process over a square mask, to investigate the general characteristics of the profile evolution. These results are shown in Fig. 2a–c and Supplementary Video 1. The simulation agrees with observation in that the ELO first appears over the sharp corners and the overall shape of the opening above the mask gradually transitions from that of the mask to a round shape as the deposited thickness increases. Similar transitions are also found in masks of other regular geometries (Supplementary Fig. 13). We also construct masks of random geometries, as shown in Fig. 2f, and Supplementary Fig. 14. As ELO proceeds, in all cases, the opening shrinks and coalesces above the mask to form a cavity between the epi-Ge and the mask. Driven by surface tension, the inner surface of the cavity continues to evolve: the hyperboloidal shape at pinch-off in Fig. 2d converts into a stable hemispherical shape in Fig. 2e. This shape evolution is consistent with the observations in Fig. 3b and c, with the exception of the surface faceting.

In the model, the directional motion of Ge atoms is driven by the negative gradient of chemical potential. In particular, it is assumed that the contribution from surface energy, $-\gamma_0\nabla^2 C$, dominates the process. This is shown most clearly in the random mask simulations in Fig. 2f and Supplementary Fig. 14. The interface advance velocity (marked by arrows in Fig. 2f$_1$–f$_3$) is mainly mediated by the Laplacian of the interface mean curvature (shown by the color scale in Fig. 2f$_1$–f$_3$), driving Ge atoms to fill the "valleys" and flatten the "peaks" to minimize interface area, similar to the dewetting phenomenon[35,36]. The excellent consistency between the simulated and experimental results (Fig. 2g and h) on anisotropic-isotropic transition and cavity formation justifies our assumption on the key role of surface energy in determining the surface profile induced by ELO. The demonstration that cavity formation is not sensitive to the shape of the mask may be advantageous in applications since it gives a large fault tolerance to the lithography process that defines the corner shape and edge roughness of the mask.

We now discuss the details of the growth front/substrate geometry during cavity formation. In the later stage of isotropic growth (Fig. 2c$_2$–d$_2$), the growth front profile exhibits an obtuse angle on the mask, leading to growth front "noses" that pinch off to form the cavity. Figures 3a–c shows the details of the experimental growth front morphology evolution on a triangular mask before and after pinch-off. In Fig. 3a, the height of the growth-front nose is denoted as $h_1$ and its profile is illustrated by the blue dotted line. By the time of Fig. 3b, the diameter of the growth-front nose decreases from ~ 1.25 μm to 0.5 μm whereas its height remains $h_1$. Moreover, the growth of the nose is accompanied by the recession of the surrounding surface (marked by blue and red dotted lines on Fig. 3a, b), clearly demonstrating long-range directional migration of adatoms as marked by the blue arrow in Fig. 3b. This phenomenon is also revealed in our simulations, where simultaneous growth and recession of the surface profile is only observed when sufficient contribution from surface diffusion is included, as indicated by the crossing of two growth-front profiles shown in Fig. 3d in a typical evolution process; no recession is observed when the surface diffusion is turned off ($M_s = 0$, Supplementary Fig. 15). Evidently, the curvature-induced surface diffusion of Ge, which accelerates the necking and pinch-off process, plays an important role in the final profile of ELO and no appreciable recession is observed at the contact line on the mask (Fig. 3b–c).

To reveal the detailed coalescence process from necking and pinch-off of the hyperboloidal surface to the formation of the dome-like cavity, we compare simulations (Fig. 3e–g) with experimental measurements (Fig. 3a–c). The progression of morphologies suggests that this topological change is also dominated by surface energy. Having a negative mean curvature (the growth front is a saddle point with positive curvature in one direction and negative in the other) and thus lower chemical potential than other regions, the neck region attracts an influx of Ge atoms (blue arrow in Fig. 3e), promoting the further constriction of the neck (its velocity is represented by black

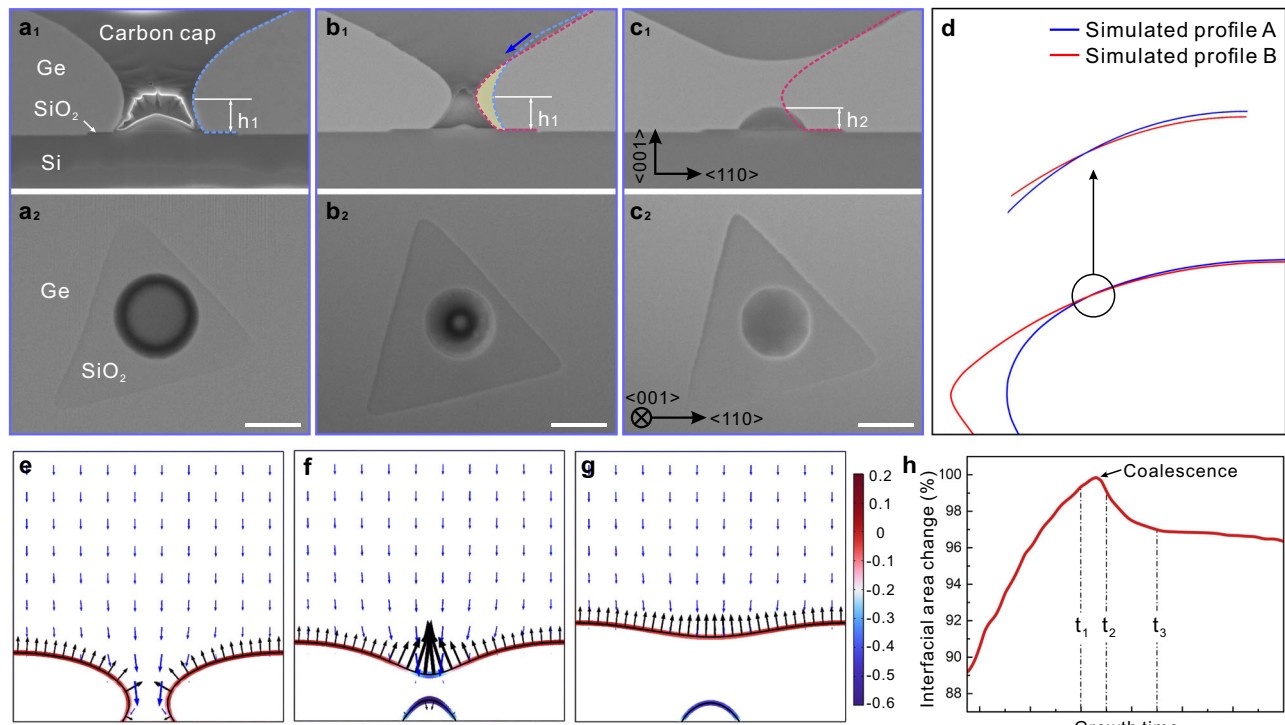

**Fig. 3 | Growth front evolution.** Upper images, $a_1$–$c_1$, are cross-sectional views at the center of a triangle mask when the growth fronts have separations of **a**, 1.25 μm, **b**, 0.5 μm, and **c**, zero (fully coalesced). Blue and red lines represent the growth front profiles. Blue arrow in $b_1$ indicates the adatom migration direction; yellow shaded regions indicate Ge adatoms that came both from the gas flux and adatom migration from upper planar facets. $a_2$–$c_2$ are corresponding plan-view SEM images showing that the Ge/mask interface changes from a round to a dodecagonal shape after coalescence ($c_2$). The scale bar is 1.0 μm. **d** Simulated evolution of the growth-front profile, at two stages before coalescence. Inset shows an enlarged view near the crossover. **e**–**g** Numerical evolution of interface during ELO. Color map indicates the normalized mean curvature. Blue and black arrows represent the flux of Ge atoms and the advancing velocity of the interface, respectively. Black curves indicate the gas-solid interface, i.e. the $C = 0.5$ contour lines. **h** Interfacial area variation approaching and after coalescence. $t_1$-$t_3$ represent the interfacial area at **e**–**g**, and the area at coalescence is used for normalization.

arrows in Fig. 3e). The contact line of the Ge surface on the mask then stalls due to the low wettability and the largely decreased flux. As a consequence of the neck constriction and contact-line stalling, the surface area continues to grow at this stage (in Fig. 3h from $t_1$ to $t_2$), while the mean curvature at the neck decreases towards a singular point. The constriction further accelerates until the ultimate pinch-off.

The system undergoes a topological change as the gaseous phase is separated into two isolated domains. On the upper surface, external incoming Ge atoms tend to migrate to the more curved region of lower chemical potential, smoothing the top surface. Inside the closed cavity, gaseous Ge atoms are soon depleted, after which the volume of the cavity becomes invariant. At pinch-off, the apex of the cavity has larger negative curvature (Fig. 3f), promoting adatom diffusion towards the apex. The flattening and rounding of the cavity (as shown in Fig. 3f, g) reduces its interfacial area ($t_2$ to $t_3$ in Fig. 3h) and minimizes the interfacial free energy. Our experiments (Fig. 3$b_1$–$c_1$) show that the height of the cavity $h_2$ indeed decreases from that of the growth front before coalescence ($h_1$), while the contact angle also decreases from 135° to 126° at the Ge/mask interface (Fig. 3b, c, Supplementary Figs. 16 and 17, and Supplementary Note 3).

We anticipate that the cavity should display facets, reflecting the orientation-dependent surface energy of crystalline Ge[37–40], and indeed faceted Ge islands in equilibrium on Si are well known[41–44]. Faceting of the cavity is visible experimentally in Fig. 3$b_2$–$c_2$, where the contact line of Ge on the mask changes from circular to dodecagonal with some edges parallel to Si <110>. Note that the faceting becomes more clearly visible in images after pinch-off, due presumably to the change in flux environment once the cavity is enclosed. The same process is observed for other mask shapes (Fig. 1c and Supplementary Fig. 18). Imaging in

cross-section along Si[110] (Fig. 4a) shows that the cavity is bounded by {111}, {311}, and (100) planar facets. We find that the orientation and spatial configuration of the cavity are unaffected by the rotation angle of the mask on Si (001) and instead the cavity shape is defined by the substrate and epilayer crystal directions (Fig. 1c and Supplementary Fig. 8). We therefore extended our model to include an orientation-dependent surface energy using the procedure developed in ref. 45. Based on surface energy calculations (Supplementary Note 4), internal configurations of the cavity are constructed and illustrated in Fig. 4b. The (001), {113}, {111}, {15 3 23}, and {20 4 23} indices, known to be common facets throughout the Ge-Si heteroepitaxial system[42], occur in our simulations with the extended model of anisotropic surface energy, consistent with our SEM (Supplementary Videos 2–3) and TEM observations. It is also worth pointing out that, in the current evolution process including the anisotropic to isotropic transition, although no dislocations were observed after coalescence (Fig. 4a and Supplementary Fig. 9), it seems likely that dislocations will result from coalescence above larger masks[9,17], especially if elongated in one direction.

## Controlling the dimension of the formed cavities
We finally consider the degree to which the cavity volume can be understood and controlled. If the dodecagonal contact line of the cavity at Ge/mask interface is approximated as a circle, its diameter is found to be a function of the dimension of the mask, as shown in Fig. 4c. Specifically, for various geometries, the diameter of the cavity increases with the mask dimension up to ~2.0 μm, beyond which a plateau is reached. Our simulation results (blue solid curve on Fig. 4c) confirm this observation: the cavity diameter increases with the enlargement of pattern size and then plateaus. The close-to-linear

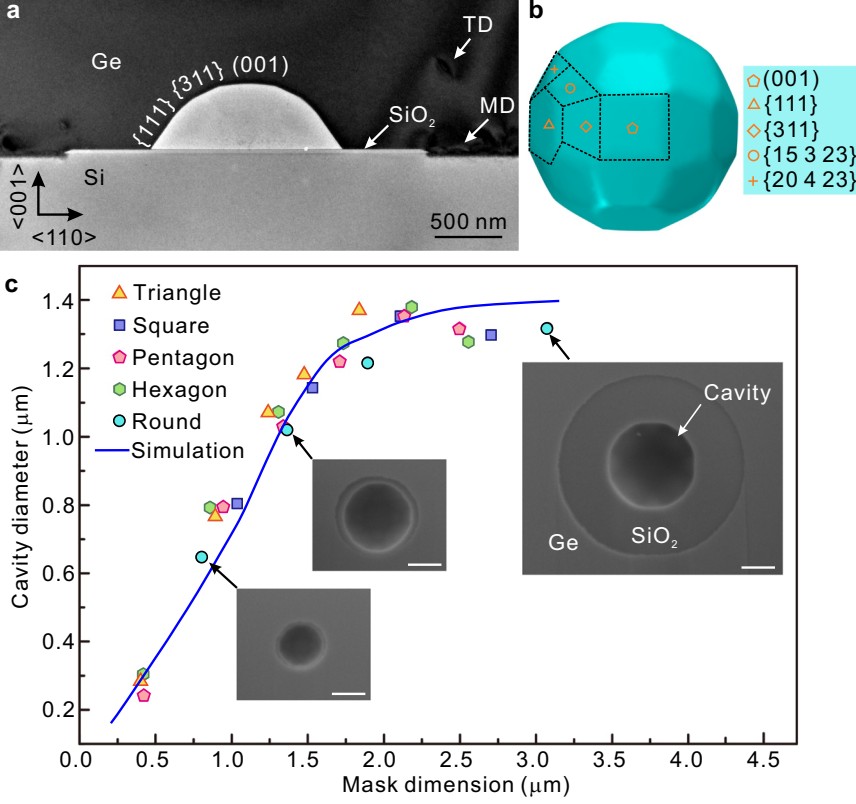

**Fig. 4 | Cavity dimension as a function of mask dimension. a** TEM image of cross-section view at the center of mask after full coalescence, demonstrating the cavity is confined by {111}, {311}, and (001) facets along <110> direction. Misfit dislocations are seen on the Ge/Si interface. **b** Simulated result of a faceted cavity through phase-field modeling, with indicated facets. The facet boundaries are sketched by dashed lines. **c** Cavity diameter as a function of dimension of the mask, using the diameter of the inscribed circle for non-circular masks. The cavity diameter increases from 0.24 μm to 1.38 μm first and then stabilizes as the dimension of the masks increases. Inset SEM images show cavities formed at the center of the round mask with different dimensions. The scale bar is 500 nm. The blue line presents simulation results on round masks upon taking the length scale $L_0 \approx 15$ nm.

relation at small dimension may be understood from an energy perspective. As discussed above, the surface energy plays a dominant role in the process. With negligible contribution from other bulk energies (particularly strain energy), surface energy alone does not yield any intrinsic length scale, aside from the less relevant interfacial thickness $L_0$ which arises from phase-field formulation. Therefore, the ELO is close to a self-similar process, and the diameter of the cavity is expected to scale with the characteristic size of the mask, the only thermodynamic length scale in the system. In this regime, i.e., diameter of mask below ~2.0 μm, larger masks take longer epi-growth time to coalesce, leading to a wider ELO-Ge in the lateral direction and higher growth front in the vertical direction, thus resulting in a larger diameter and volume of the cavity. The size of the cavity can thus be tuned by the dimension of the mask.

Although the system contains no intrinsic length from thermodynamics, the kinetic parameters lead to a characteristic length. More specifically, a comparison between the effects of surface diffusion and bulk diffusion in the gas phase leads to another length, $L_s = \sqrt[3]{M_s \eta / I}$, which characterizes the contribution of surface diffusion of Ge adatoms relative to the deposition flux (Supplementary Note 5). Due to the much slower diffusion of Ge in the solid phase, the Ge atoms are immobilized once deposited, except those on the surface that can migrate via surface diffusion. In a system smaller than $L_s$, surface diffusion dominates, and the Ge adatoms undergo substantial migration to reach a close-to-equilibrium state of the system. The surface diffusion, however, is less effective in a larger system. In a system larger than $L_s$, with the surface diffusion-kinetics ineffective, the ELO-induced topography deviates from the thermodynamic equilibrium state, and

thus the scaling in the mask dimension may not hold. Experiments involving masks larger than ~2.0 μm show that the growth fronts experience a self-similar evolution process in the early stage, dominated by the uniform deposition flux, and less affected by the size-dependent surface diffusion. Thus, the growth front morphology hardly varies (Supplementary Fig. 19), leading to a fixed cavity size at coalescence, independent of the mask size under current growth conditions. Such a phenomenon could not be captured in a conventional phase-field model without accounting for the effect of surface diffusion.

## Summary and outlook

In summary, we have described a general phenomenon of Ge epitaxial layer overgrowth over a dielectric mask in which the growth front approaches a universal isotropic shape and closes to form a single crystalline confined cavity; we have shown control of dimension and position via mask parameters, and presented a modeling strategy for this three-dimensional process that enables design of structures combining single crystals with controlled voids.

The phenomenon by which cavities can be formed in thick, single-crystal epitaxial Ge layers on Si uses isolated masks to control the initial deposition sites during a gas-phase growth process. A combination of microscopy and phase-field modeling demonstrates that the formation of cavities proceeds through propagation of a three-dimensional growth front inwards from the mask edges, followed by coalescence at some height above the surface to enclose the cavity. This process appears to be controlled by surface diffusion of Ge atoms driven by the net negative curvature of the surface

at the growth front. As the growth front propagates, it advances more quickly at regions of higher curvature, driving a transition from anisotropic to isotropic morphology that is overall insensitive to the mask shape and eventually pinches off the cavity. The cavity shape is determined by surface energy minimization after coalescence, while the cavity size depends on mask size, if the mask is sufficiently small. This demonstration and modeling of three-dimensional morphology formation is distinct from prior experiments where elongated cavities form after ELO[7–12] or are defined lithographically[13,15,16], and offers new opportunities for the growth of single crystal semiconductor materials and manipulation of their properties. The process we have described is CMOS-compatible and can generate ordered arrays of open pockets and cavities with controllable diameters. Given the dimensions in the range of hundreds of nanometers, this suggests possibilities for interdisciplinary research for electronic and optoelectronic devices with high light output power or photonic-crystal resonators for dense LiDAR (light detection and ranging) applications[13–16]. For example, with patterned AlN on sapphire substrates with regular hexagonal holes, devices fabricated based on this coalesced AlN showed an improved light output power (53.1%) compared to those without coalesced AlN, because of the reduced dislocation densities[42]. We anticipate that in Ge/Si, the morphology and dimensions can be controlled by growth parameters, while similar behaviors may occur in other group IV and III-V semiconductor[17] systems under conditions where surface diffusion dominates growth. We anticipate that a deeper understanding of the dynamics of ELO on suitably patterned substrates will help to work towards precise control of spontaneous evolution in epitaxy processes, and will enable fabrication of new types of devices that make full use of the possibilities for engineering materials properties.

## Methods
### Wafer cleaning and lithography
8-inch Si (001) wafers (p-type, 0.5-100 Ω cm from Resemi, Suzhou, China) with a thermal oxide of 40 nm were used in our experiments. Photoresist (SPR700) was coated and lithography (Nikon S204) was conducted to define the designed patterns. Once the developing process was finished, the wafers were transferred into a tank for buffered oxide etch (BOE, 7:1 for $H_2O$: HF) to define the patterns through oxide etching. Before loading to the reactive tube for Ge epi-growth, the wafers went through a modified Radio Corporation of America (RCA) cleaning consisting of

1. Organic/particle clean: $NH_4OH$:$H_2O_2$:$H_2O$ (1:1:5) at 80 °C for 10 minutes
2. Ionic clean: HCl:$H_2O_2$:$H_2O$ (1:1:6) at 80 °C for 10 minutes
3. Chemical oxide pattern and passivation: HF:$H_2O$ (1:50) for 15 seconds.

### Ge growth
Ge epi-growth was conducted in a commercialized ASM Epsilon 2000. The loading gas is $GeH_4$ and HCl with $GeH_4$:HCl = 10:1. To avoid Stranski-Krastanov growth[44], a two-step growth process approach is employed. A low temperature Ge buffer is first grown at 380 °C for ~60 nm. The buffered Ge plastically relaxes the majority of the strain arising from the lattice mismatch (4.2%) between the Ge film and the Si substrate[24]. A high-temperature growth is then conducted at 730 °C to achieve Ge with higher quality while providing high deposition rate as well as allowing sufficient surface diffusion. In this growth condition, the Ge deposition on patterned Si (001) surface is selective: no nucleation could be measured on the oxide mask. For both low and high temperature growth, the pressure was maintained at $2.0 \times 10^{-2}$ mbar. At this partial pressure.

## Analytical methods
To quantify the coalescence process, we terminate the growth at different stages before and after the coalescence. Top-view images of the samples were obtained by scanning electron microscopy (SEM; Zeiss Merlin). Focused ion beam preparation (FIB, Helios Nanolab 600i, FEI) was adapted to investigate the geometric structure underneath. Two effective techniques in dual FIB/SEM system are described in detail here. The first enables a cross-sectional view. Here, carbon was deposited on Ge in order to prevent surface damage by the $Ga^+$ ion beam. FIB was then used to mill the sample gradually in the Si $\langle 1\bar{1}0 \rangle$ direction and images were recorded in situ by SEM at angle of 52° (for example Fig. S6a). The second method is a lift-out technique that was employed to reveal the interior structure in plan-view (Figs. S6b–S8) as well as allowing for TEM imaging (Fig. 4a). As shown in Fig. S7, a small specimen containing Si, the oxide pattern and Ge was cut, lifted out using a tungsten probe and transferred to a copper grid. The grid was placed horizontally on flat SEM holder so that the Si-Ge interface was perpendicular to the $Ga^+$ ion beam. The grid was then rotated 90° and placed on the grid holder so the Si-Ge interface was parallel to the ion beam. Si was FIB milled step by step until the oxide pattern was exposed to reveal the cavity-on-oxide configuration. The oxide was not fully removed in order to preserve the relative positions of cavity and oxide patterns. All the TEM samples were milled and polished to ~100-nm thickness by FIB. The microstructures were analyzed by transmission electron microscopy (TEM, Tecnai F30, FEI) operated at 300 kV.

## Data availability
The data that support the findings of this study are available from the corresponding authors upon request. Source data are provided with this paper.

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

## Acknowledgements

This work is financially supported by Guangdong Provincial Innovation and Entrepreneurship Project (No. 2017ZT07C071), National Natural Science Foundation of Guangdong (No.2022A1515010216), National Natural Science Foundation of China (Nos. 52172314, 11932009), the Shenzhen Science and Technology Innovation Commission (Nos. JCYJ20210324105402007, ZDSYS20210623092005017). The Southern University of Science and Technology Core Research Facilities is acknowledged for providing the FIB and TEM imaging facility.

## Author contributions

R.-T.W. initiated the project and supervised the work. Y.Z. and B.W. conducted the experiments. C.M. and H.C. performed simulations. R.-T.W., W.H., and F.M.R. developed the mechanism analysis. All authors co-wrote the draft, discussed the results, and contributed to the final manuscript.

## Competing interests

The authors declare no competing interests.
