## [Peer Review File · Nature Communications]

Reviewers' Comments:

Reviewer #1:

Remarks to the Author:

The paper is focussed to the morphologic and modelling issues, which cannot be considered any breakthrough. However, the experimental result of having a freestanding cap of Ge on the oxide pad, which is free of dislocations, is very interesting, original and promising. Therefore, I suggest the authors to move to the main text and enrich by further measurements (and statistics) the characterization of the freestanding cap, whereas part of the modelling can surely be placed in the supplementary material. To this regard, it is not even clear how the authors could obtain a downward faceted cap if the Phase Field method does not include anisotropies in surface energy (as far as I understand).

Reviewer #2:

Remarks to the Author:

The authors reported an interesting phenomenon in the lateral growth of Ge on Si, in which cavities are formed on SiO₂ masks. The authors presented not only experimental results (mask shape/size dependence, time evolution, SEM with FIB etching, etc.) but also detailed theoretical analysis. The results are valuable, but I have following concerns and suggestions for revisions.

(1) A careful discussion of dislocation-free Ge at the coalescence point should be provided, given that dislocations can be induced at the coalescence point by slight misalignments in the crystallographic direction of laterally grown Ge. See, for example, IEEE JSTQE 24, 8201007, 2018, where the SiO₂ mask was a line-and-space pattern.

(2) The final sentence in the abstract "we suggest that this mechanism provides a strategy for developing electronic and photonic functionalities" should be provided more explicitly in the text. I do not see any such discussion.

(3) I am not sure about a2 - e2 in Fig. 2; it would be easier to understand if the Ge layers were shown in blue, as in a1 - e1. According to the video data in the supplementary material, the top surface shape was supposed to be drawn by integrating in the in-plane direction. A simple cross-section at the cavity center would be more effective.

(4) Effect of the growth conditions (temperature, growth rate, reactor pressure, etc.) would be discussed, because the growth at a relatively high temperature should prefer the round-shape, as observed in this work, rather than the faceted one. A discussion would be also included on the result that the Ge layer showed a tendency in contact with the SiO₂ mask, but this is against the selective growth, where Ge do not prefer a contact with the SiO₂ mask.

(5) The CVD growth was probably performed in a viscous flow regime according to the theoretical modelling, rather than a molecular flux regime. However, I cannot find the condition such as the gas pressure of the CVD reactor. This information is needed for the theoretical modeling and analysis.

Reviewer #3:

Remarks to the Author:

The manuscript reports on the epitaxial lateral overgrowth of Ge on a masked Si substrate. The manuscripts contains high-quality results, and is well written. Besides some technical comments, which are detailed below, I'm mainly concerned about the novelty of the work. ELO (or LEO) has been studied in detail for a large class of materials (see for instance this review: <https://doi.org/10.1016/j.pquantelec.2021.100316>). Coalescing crystals on a mask have been studied and the formation of air gaps/voids or cavities has been reported. The authors should thus make clear what is the state-of-the-art, and what are the novel aspect of the current manuscript.

On page 4 the anisotropic-isotropic transition is mentioned. This transition is later discussed in the theory section. It would be nice to forward refer to that.

It is not clear how these cavities could be useful for any photonic or electronic device. Would be nice if this could be worked out in more detail.

Reviewer #4:

Remarks to the Author:

This is a very nicely written, clear paper. My recommendation is to publish with only minor suggested revisions.

As a non-theorist, I can't comment on the details of the theory, but found the associated text explanations to be helpful, logical, and consistent with research on a wide range of different growth systems. Comments 1-5 are minor suggested revisions/clarification. Comments 6-8 are follow-on thoughts (and not criticisms), intended more as guidance from a test-reader for things could perhaps be added to this paper if it can be easily/logically done, or considered for follow-on research or publications:

(1) Comment on TDs: If the methods were applied to a lattice-matched system (Ge grown on a Ge substrate), then no TDs would be expected. For Ge-on-Si, MDs and TDs are expected (and one is seen in Figure 4a). Although this paper does not claim to reduce the TDD, it comes close to suggesting it in line 43 of the introduction. It might be worth clarifying that any TD "filtering" in this system would be related to two or more TDs entering a cavity, and fewer cavities exiting it. This would be statistically more likely to happen at a high TDD, and less likely at a low TDD. This clarification would not detract from the paper, which is primarily about mechanisms of void formation which are extensible to many material systems. My motivation for making this recommendation is that the authors seem to understand that SAE in and of itself does not eliminate TDs from lattice-mismatch, and have an opportunity with this paper to help address this common misconception. Perhaps this point could easily be made after line 97, by pointing out that TDs associated with MDs are expected and observed, as seen in Figure 4a.

(2) Figures 2(a2-e2) were initially very confusing, because I thought the aqua color was a cross-section through something. Perhaps consider shading the bulk Ge epilayer so that it is distinct from both the surface and the overlying gas phase. Then it might be more intuitively clear that the aqua color is the epilayer surface (and not a cross-section).

(3) Line 290: Should this be "could be captured" (delete "not")?

(4) Line 28: "patterning"

(5) Does the model include/allow for Ge to land on the masked area, then diffuse laterally to the Ge epilayer edges? Or is the Ge sticking coefficient on the mask equal to zero? Is the answer to this temperature-dependent?

(6) Is the solid-gas gradient an essential feature of the model, or could the system be modeled with only lateral surface diffusion, with a fixed flux of Ge impinging upon the surface everywhere?

(7) Can the model be extended to include anisotropic diffusion (due to Schwoebel-Ehrlich barriers)? See for example Journal of Crystal Growth 269, p276 (2004), Fig A-C. Could anisotropic diffusion (preferentially up/away from a hole) hinder/prevent coalescence?

(8) At lower growth temperatures, could the formation of low-energy facets hinder or prevent coalescence? An example of this might be Figure 3b of Crystal Growth and Design 21, p5955 (2021), although this figure is for a 1D system which is qualitatively different.

Again, this is a well-written paper which should be

Dear Editor and Reviewers,

We greatly appreciate your time in reading and commenting on our manuscript and in particular the insightful and constructive feedback of the reviewers. We feel that the reviewers' input has enabled us to produce a greatly improved manuscript. Below is our point-by-point reply (in blue) to the reviewers' comments and questions (in black). Based on these comments, we made alterations and clarifications in the main text and supporting information that are indicated with yellow highlights in marked versions of the documents. We hope that these have answered the concerns and questions raised.

Sincerely,

Rui-Tao Wen, Associate Professor, Southern University of Science and Technology (SUSTech)

Wei Hong, Professor, Southern University of Science and Technology (SUSTech)

Frances M. Ross, TDK Professor, Massachusetts Institute of Technology (MIT)

on behalf of all authors.

Response to reviewers' comments

Reviewer #1

General comments: The paper is focused to the morphologic and modelling issues, which cannot be considered any breakthrough. However, the experimental result of having a freestanding cap of Ge on the oxide pad, which is free of dislocations, is very interesting, original and promising. Therefore, I suggest the authors to move to the main text and enrich by further measurements (and statistics) the characterization of the freestanding cap, whereas part of the modelling can surely be placed in the supplementary material. To this regard, it is not even clear how the authors could obtain a downward faceted cap if the Phase Field method does not include anisotropies in surface energy (as far as I understand).

Response: We appreciate the reviewer's comment that *"the experimental result of having a freestanding cap of Ge on the oxide pad, which is free of dislocations, is very interesting, original and promising."*

The reviewer asks for additional measurements of the freestanding cap. We agree that the promising nature of this aspect of the cavity growth warrants further description. In particular, we present additional images showing the lack of observed dislocations at the coalescence point. To improve our confidence in the result we based this finding on more than 10 TEM samples. In **Fig. R1** below (now added into **Fig. S9**), the TEM images in cross-section and plan-view showed no observable dislocations at the coalescence point, independent of the shape of the oxide mask. In particular, in **Fig. R1c** the TEM lamella was taken just above the coalescence point and showed no visible dislocation formation.

To emphasize the statistical robustness of the observations, for each dielectric mask with varied shape and size we checked at least 3 samples to confirm the diameter and profile of the formed cavities to generate the results reported in the manuscript. We also measured the dislocation density between the mask regions, $2 \times 10^7 \text{ cm}^{-2}$. This value is consistent with our previous measurements from standard vertical epi-Ge on Si (*J. Appl. Phys.* 133, 075703 (2023)), so we therefore assign the observed dislocations at the regions away from the masks to the threading dislocations originating at the Si/Ge interface.

We added this information on p. 5 in the revised version: *"To verify these results, we checked at least 3 samples for each mask shape and size, which yielded consistent diameters and profiles for the formed cavities. An important feature of the growth process is that the ELO Ge above the cavity is found to be dislocation-free at the coalescence point, based on statistical information from over 10 TEM samples with different mask shapes (Fig. S9). Between mask regions, threading dislocations are observed at about $2 \times 10^7 \text{ cm}^{-2}$ (a value consistent with our previous measurements from vertical epi-Ge on Si (*J. Appl. Phys.* 133, 075703 (2023))) that are associated with misfit dislocations originating at the Ge/Si interface (Fig. S9). The ELO Ge over the mask therefore offers, as a byproduct of the lateral growth and coalescence, an opportunity to achieve dislocation-free Ge, at least in microscale regions."*

Fig. R1 (a-b) Dark field TEM cross-sectional view across the mask after full coalescence. **c-d,** Plan-view TEM lamella from circular and pentagonal oxide pads. The lamellae were fabricated above the cavity, with plane parallel to Si (001). The FIB milling process was from the Si to Ge, so that the position of the mask can be seen. This is marked with white dashed outlines. No threading dislocations were observed at the coalescence point (the center of the mask).

In terms of the modeling of faceting by the phase field method, we agree that conventional phase field methods do not include anisotropy in surface energy. Therefore, to account for anisotropy and to model the faceted surface, we replaced the constant surface energy, γ , with an orientation-dependent term (Supplementary note 4):

$$\gamma(\mathbf{n}) = \gamma_0 \left[1 - \sum_{i=1}^M \beta^i (n_j q_j^i)^{w^i} H(n_k q_k^i) \right]. \quad (\text{R1})$$

where $\mathbf{n} = \nabla C / |\nabla C|$ is the unit normal vector of the solid-gas interface, the dummy variables j and k indicate Einstein summation, \mathbf{q}^i is the unit normal vector of one of the M facet surfaces, β^i and w^i are material-specific parameters, helping control the width and depth of the corresponding energy well, and $H(x)$ is the Heaviside function.

For clarification, we have modified the sentences on p.6, after Eq. (1), for clarification and to direct the readers to the supplementary note: “*Here, an isotropic and orientation-independent surface energy is adopted to reduce computational cost in most numerical simulations, except for the simulations showing a faceted surface. These are obtained through an extended model in which the constant surface energy γ is replaced by an orientation-dependent term, as detailed in Supplementary Note 4. Numerical tests have not shown significant changes in the modeled phenomena, except for the faceted surface at equilibrium.*” Also on p.13, in the discussion related to **Fig.4b** (the only numerical result shown with a faceted surface), we modified the sentence to “*Based on surface energy calculations (Supplementary note 4), internal configurations of the cavity are constructed and illustrated in Fig. 4b. The (001), {113}, {111}, {15 3 23} and {20 4 23}*”

indices, known to be common facets throughout the Ge-Si heteroepitaxial system³⁶, occur in our simulations with the extended model of anisotropic surface energy, consistent with our SEM (movies S2-S3) and TEM observations.”

Reviewer #2:

General comments: The authors reported an interesting phenomenon in the lateral growth of Ge on Si, in which cavities are formed on SiO₂ masks. The authors presented not only experimental results (mask shape/size dependence, time evolution, SEM with FIB etching, etc.) but also detailed theoretical analysis. The results are valuable, but I have following concerns and suggestions for revisions.

Response: We thank the reviewer for the comment on “*an interesting phenomenon*”, with “*not only experimental results but also detailed theoretical analysis*”. We address the concerns and suggestions below.

Comment 1: A careful discussion of dislocation-free Ge at the coalescence point should be provided, given that dislocations can be induced at the coalescence point by slight misalignments in the crystallographic direction of laterally grown Ge. See, for example, IEEE JSTQE 24, 8201007, 2018, where the SiO₂ mask was a line-and-space pattern.

Response: We agree that the lack of dislocations observed at the coalescence point is an interesting and surprising aspect of our experiments. We have deepened our discussion of this result through additional images and text as described in the response to Reviewer 1, in particular discussing the fact that the result is based on imaging multiple samples. The paper mentioned above uses an array of long dielectric masks, rather than the smaller isolated masks used in our work. (The strip width was comparable to our mask diameters.) It is possible that the differences in dislocation density may be due to the different misalignments that may result from coalescence of the two long growth fronts on either side of the strips in IEEE JSTQE 24, 8201007, 2018, versus coalescence of a growth front (including the anisotropic-isotropic transition we described) at the center of our isolated masks. The possible relationship between mask shape and probability of dislocation is interesting and worth further study in the future and we appreciate the reviewer pointing out this possibility. **We have cited the two papers** (IEEE JSTQE 24, 8201007 (2018) and *Progress in Quantum Electronics* 77 100316 (2021)) **in added text on page 13, “it seems likely that dislocations will result from coalescence above larger masks, especially if elongated in one direction.”**

Comment 2: The final sentence in the abstract “we suggest that this mechanism provides a strategy for developing electronic and photonic functionalities” should be provided more explicitly in the text. I do not see any such discussion.

Response: Thank you for this important suggestion. Fabricating a cavity-based device is challenging for us but existing literature suggests applications of the type of structure we demonstrated. An example is Shen et al. (Nature Materials, 22, 853–859 (2023)) where, by adopting patterned AlN on sapphire substrates with regular hexagonal holes (**Fig. R2a**), the ELO AlN after coalescence yields a low dislocation density of $3.3 \times 10^4 \text{ cm}^{-2}$. Devices fabricated based on this coalesced AlN showed an improved light output power (53.1%) compared to those without

coalesced AlN. Another example is the air cavities implemented in p-AlGaAs to form double-lattice photonic-crystal resonators (**Fig. R2b**), achieving a brightness of over $300 \text{ MW cm}^{-2} \text{ sr}^{-1}$ Noda *et al.* (Nature Materials 18, 121–128 (2019)). Such a cavity based double-lattice photonic-crystal resonator is promising for developing dense LiDAR (light detection and ranging) applications. **We have discussed these two demonstrations and cited the two papers (refs. 41 and 42) in the “Summary and outlook” on p.16.**

Fig. R2 a, Cross-sectional dark-field TEM images of AlN on sapphire substrates under two-beam conditions with $g = [0002]$ and $g = [11\bar{2}0]$. Images adapted from *Nature Materials*, 22, 853–859 (2023). **b**, Top and cross-sectional SEM images of the fabricated double-lattice photonic crystal. Images adapted from *Nature Materials* 18, 121–128 (2019).

Comment 3: I am not sure about a2 - e2 in Fig. 2; it would be easier to understand if the Ge layers were shown in blue, as in a1 - e1. According to the video data in the supplementary material, the top surface shape was supposed to be drawn by integrating in the in-plane direction. A simple cross-section at the cavity center would be more effective.

Response: We agree that the figure was not clear. In **Fig. 2a-2e** in the previous version, the cyan-blue represents the interface between the solid and gas Ge. As suggested, we have shown the cross-section view from the cavity center in the revised manuscript. We also point out that in **Fig. 2g**, the solid Ge phase is removed to reveal the profile of the oxide.

Comment 4: Effect of the growth conditions (temperature, growth rate, reactor pressure, etc.) would be discussed, because the growth at a relatively high temperature should prefer the round-shape, as observed in this work, rather than the faceted one. A discussion would be also included on the result that the Ge layer showed a tendency in contact with the SiO₂ mask, but this is against the selective growth, where Ge do not prefer a contact with the SiO₂ mask.

Response: Thank you for this suggestion to expand the discussion of growth parameters. We agree that at a lower growth temperature, for example, 450°C, the mobility of the Ge atoms is relatively low. Thus, it is difficult to even form a smooth profile of the epi-Ge. For example, we conducted epi-Ge growth on a patterned dielectric mask with a square shape. The results are shown in **Fig. R3** below (now added to the revised supplementary information as **Fig. S2**). The Ge forms a very rough

surface with deep grooves on the {113} facets (**Fig. R3b-c**), presumably because of low Ge mobility and perhaps sensitivity to imperfections of the oxide strips defined by lithography (**Fig. R3e**). Therefore, we did not use this growth condition for constructing the cavities. Furthermore, at lower temperature, the contact angle between Ge and SiO₂ is over 90°, different from the acute angle as shown in **Fig. 3**.

The lower temperature, on the other hand, is not expected to change the thermodynamics of the system, and the interfacial energy between Ge and SiO₂ is still much higher than that of the Ge-Si interface. Therefore, despite the lower mobility, the high energetic driving force is still able to deplete the Ge originally deposited over the energetically unfavorable regions of the SiO₂ mask.

To explain our choice of growth conditions more clearly in the manuscript, we added text on p. 4: “Growth at high temperature yields both a high deposition rate and high mobility of Ge adatoms. Lower temperature growth (see Fig. S2) does not produce favorable morphologies because of the low mobility of Ge adatoms. Instead the growth front is irregular with an obtuse angle (an acute angle is a prerequisite to form a cavity) and overgrowth to an anisotropic-isotropic transition was not visible.”

Fig. R3 Low temperature growth of Ge on patterned Si (001). **a**, The mask used, consisting of 20×20 μm² square array of oxide strips of width 1.0 μm. **b**, Top view SEM image of this pattern after epi-growth. Growth formed a trapezoid shape with dominant facets {311} but with rough profile, presumably due to the low mobility of Ge adatoms during growth. **c**, Tilted SEM view of the low temperature epi-Ge after carbon deposition for FIB milling. Surface grooves can be clearly seen at these growth conditions. **d**, tilted view of the oxide stripe, showing the oxide covered by Ge without forming a tunnel or cavity. **e**, View of the edge of the lithographically defined mask. Imperfections can be observed which may contribute to the Ge growth irregularities due to low mobility.

Comment 5: The CVD growth was probably performed in a viscous flow regime according to the theoretical modelling, rather than a molecular flux regime. However, I cannot find the condition such as the gas pressure of the CVD reactor. This information is needed for the theoretical modeling

and analysis.

Response: For both low and high temperature growth, the pressure was kept at 2.0×10^{-2} mbar. We agree that it is unclear whether the transport of Ge should be regarded as diffusion or convection (flow). Fortunately, the two processes follow similar mathematical equations, except that a flow is often driven by pressure gradient rather than chemical potential. In the model presented in this manuscript, our aim is to demonstrate the governing mechanism of the morphological evolution, rather than focusing on the origin of the Ge flux. For this purpose, the effective kinetic parameter related to Ge transport is the deposition flux, prescribed as a boundary condition for the diffusion equation for simplicity. A pressure boundary condition is incompatible with our diffusion formulation of the deposition process. We would not expect any substantial change if we regarded this as the flow rate of an equal amount of Ge atoms, if we switch the formulation into a pressure-driven flow. **We added this information “For both low and high temperature growth, the pressure was maintained at 2.0×10^{-2} mbar.” into the Materials and Methods in the supplementary information in p. 3 and “Here, for consistency with that in the solid phase, the transportation of Ge atoms in the gas phase is also modeled as a diffusion process. As the focus of the current model is the morphological evolution of the solid phase, the actual transportation process of Ge in the gas phase is deemed to be a minor factor, as long as the deposition flux is properly prescribed and the topology of the gas domain accounted for.” on p.7, after Eq. (4).**

Reviewer #3:

General comments: The manuscript reports on the epitaxial lateral overgrowth of Ge on a masked Si substrate. The manuscript contains high-quality results, and is well written. Besides some technical comments, which are detailed below, I’m mainly concerned about the novelty of the work. ELO (or LEO) has been studied in detail for a large class of materials (see for instance this review: <https://doi.org/10.1016/j.pquantelec.2021.100316>). Coalescing crystals on a mask have been studied and the formation of air gaps/voids or cavities has been reported. The authors should thus make clear what is the state-of-the-art, and what are the novel aspect of the current manuscript.

Response: We appreciate the reviewer’s positive comments that “*The manuscript contains high-quality results, and is well written.*”. We agree that ELO has been studied by other groups and that it is important to distinguish our work from prior experiments.

The mentioned review paper (Progress in Quantum Electronics 77 100316 (2021)) deals with the ELO of III-V semiconductor materials, and most of the dielectric masks were designed to be a linear array of strips. Under these conditions ELO of III-V materials forms tunnels when growth fronts meet along the short dimension of the strip, but the long coalescence front leads to unpredictable coalescence, as shown in the figures of this review paper. In our work for Ge on Si, the anisotropic-isotropic transition is independent of the shape of our dielectric masks, with coalescence isotropic and dominated by surface energy, forming a small cavity that can be well controlled in terms of size and crystal quality after coalescence.

In comparison with this and other prior work, the uniqueness of our results is that:

- 1: We describe a general phenomenon of Ge epitaxial layer overgrowth over a dielectric mask in which the growth front approaches a universal isotropic shape and closes to form a single crystalline confined cavity;
- 2: Control of dimension and position via mask parameters, hence opportunities to generate arrays of open pockets or cavities with potential electronic and optoelectronic applications;

3: A modelling strategy for this three-dimensional process that enables design of structures combining single crystals with controlled voids.

To clarify these three novel aspects that distinguish our work we have changed the text on p 16. We cite this paper (now reference 7 in the revised version) in the introductory paragraph where we envisage the potential of this anisotropic-isotropic transition in III-V systems.

Comment 1: On page 4 the anisotropic-isotropic transition is mentioned. This transition is later discussed in the theory section. It would be nice to forward refer to that.

Response: We thank the reviewer for this suggestion and added the following sentence in the revised manuscript *“Theoretical models, developed below to understand the underlying mechanisms, have successfully reproduced these observations.”* on p. 4

Comment 2: It is not clear how these cavities could be useful for any photonic or electronic device. Would be nice if this could be worked out in more detail.

Response: Thank you for pointing this out. Please see our response to **Comment 2** of **Reviewer #2**.

Reviewer #4:

General comments: This is a very nicely written, clear paper. My recommendation is to publish with only minor suggested revisions.

Response: Thank you for this encouraging recommendation.

As a non-theorist, I can't comment on the details of the theory, but found the associated text explanations to be helpful, logical, and consistent with research on a wide range of different growth systems. Comments 1-5 are minor suggested revisions/clarification. Comments 6-8 are follow-on thoughts (and not criticisms), intended more as guidance from a test-reader for things could perhaps be added to this paper if it can be easily/logically done, or considered for follow-on research or publications:

Comment 1: Comment on TDs: If the methods were applied to a lattice-matched system (Ge grown on a Ge substrate), then no TDs would be expected. For Ge-on-Si, MDs and TDs are expected (and one is seen in Figure 4a). Although this paper does not claim to reduce the TDD, it comes close to suggesting it in line 43 of the introduction. It might be worth clarifying that any TD "filtering" in this system would be related to two or more TDs entering a cavity, and fewer cavities exiting it. This would be statistically more likely to happen at a high TDD, and less likely at a low TDD. This clarification would not detract from the paper, which is primarily about mechanisms of void formation which are extensible to many material systems. My motivation for making this recommendation is that the authors seem to understand that SAE in and of itself does not eliminate TDs from lattice-mismatch, and have an opportunity with this paper to help address this common misconception. Perhaps this point could easily be made after line 97, by pointing out that TDs associated with MDs are expected and observed, as seen in Figure 4a.

Response: Thank you for your suggestion. We agree that a byproduct of the formation of these cavities is to create a region of low dislocation density in this materials system in which dislocations are formed due to lattice mismatch. As mentioned in the response to Reviewer 1, we find the

dislocation density at regions away from the masks to be as expected from previous Ge growth on Si. We modified the text on p. 5 to specify that this TD is as expected and that the masks create regions of lower density. *“An important feature of the growth process is that the ELO Ge above the cavity is found to be dislocation-free at the coalescence point, based on statistical information from over 10 TEM samples with different mask shapes (Fig. S9). Between mask regions, threading dislocations are observed at about $2 \times 10^7 \text{ cm}^{-2}$ (a value consistent with our previous measurements from vertical epi-Ge on Si (J Appl Phys 133, 075703 (2023))) that are associated with misfit dislocations originating at the Ge/Si interface (Fig. S9). The ELO Ge over the mask therefore offers, as a byproduct of the lateral growth and coalescence, an opportunity to achieve dislocation-free Ge, at least in microscale regions.”*

Comment 2: Figures 2(a2-e2) were initially very confusing, because I thought the aqua color was a cross-section through something. Perhaps consider shading the bulk Ge epilayer so that it is distinct from both the surface and the overlying gas phase. Then it might be more intuitively clear that the aqua color is the epilayer surface (and not a cross-section).

Response: Thank you for your suggestion. We have made the revisions as suggested. See the revised Fig. 2a₂-e₂.

Comment 3: Line 290: Should this be "could be captured" (delete "not")?

Response: Thank you for capturing the typo here which caused a logic error. We have modified the sentence in the revised manuscript as: *“Such a phenomenon could not be captured in a conventional phase-field model, without accounting for the effect of surface diffusion.”*

Comment 4: Line 28: “patterning”

Response: Thank you for pointing this out, we have corrected the typo.

Comment 5: Does the model include/allow for Ge to land on the masked area, then diffuse laterally to the Ge epilayer edges? Or is the Ge sticking coefficient on the mask equal to zero? Is the answer to this temperature-dependent?

Response: Yes, the model does allow landing of Ge on the SiO₂ masked area. The difference in free energy then drives the Ge to diffuse laterally away from the masked area. In fact, the competition between the Ge-SiO₂ interface energy and that of the free Ge surface will lead to the partial coverage of the masked region. Without experimental data on the temperature dependence of the corresponding energies, we take them to be temperature independent in the analysis.

For clarification, we have added the following statement in the revised manuscript, page 8:

“Such a boundary condition does not prevent Ge adatoms landing on the SiO₂ surface, but effectively increases the free energy by enforcing a solid-gas interface with the prescribed contact angle.”

Comment 6: Is the solid-gas gradient an essential feature of the model, or could the system be modeled with only lateral surface diffusion, with a fixed flux of Ge impinging upon the surface everywhere?

Response: Indeed, the diffuse interface is an essential feature of the phase field model. While a sharp interface model can, in principle, capture the morphological evolution by tracking the

deposition flux on the surface and allow lateral diffusion, it is cumbersome to deal with the case of topological change, e.g. the coalescence of a cavity.

For clarity, we have added the statement:

“In contrast to a sharp interface model, a phase field model does not require explicitly tracking the motion of interfaces, and thus facilitates modeling of complex morphological and even topological evolution.” on page 6.

Comment 7: Can the model be extended to include anisotropic diffusion (due to Schwoebel-Ehrlich barriers)? See for example Journal of Crystal Growth 269, p276 (2004), Fig A-C. Could anisotropic diffusion (preferentially up/away from a hole) hinder/prevent coalescence?

Response: Yes: the model leaves a large margin for further modification based on experimental observations, e.g. by modifying the mobility tensor to include anisotropic diffusion. However, the coalescence of a cavity is dominated by the energetics, i.e. the minimization of surface energy. We therefore expect that introducing anisotropic diffusion alters the kinetic process, and may delay the coalescence, but cannot prevent it from happening.

This is an interesting point to explain to the readers and we have added the following discussion to Supplementary note 1 on page 6 of the Supplementary Materials:

“It is also possible to extend the current model to include anisotropic diffusion (which might originate from effects such as an Ehrlich-Schwoebel barrier (J. Chem. Phys., 44 (1966), 1039; J. Appl. Phys., 40 (1969), 614.), by introducing orientation dependence in the already anisotropic mobility tensor M in Eq. (S5). Anisotropic diffusion changes the kinetics by directing adatom flux along preferred directions, but we expect that since it does not alter the thermodynamics, it may delay some phenomena such as coalescence, but would not prevent them from happening.”

Comment 8: At lower growth temperatures, could the formation of low-energy facets hinder or prevent coalescence? An example of this might be Figure 3b of Crystal Growth and Design 21, p5955 (2021), although this figure is for a 1D system which is qualitatively different.

Response: At lower growth temperatures, such as 450°C, we found that coalescence to form cavities does not take place. However, we believe this is because of low mobility at low temperature rather than a result of the specific facets present. Please see the description in our response to **Comment 4 of Reviewer #2**. At low temperature, cavity formation failed due to the irregular growth front with an obtuse angle; overgrowth to an anisotropic-isotropic transition was not visible. A growth front with an acute angle is a prerequisite to form a cavity. In the paper mentioned by the reviewer, involving GaAs ELO over strip masks aligned parallel to $[\bar{1}10]$, growth fronts with an acute angle coalesced along GaAs $[\bar{1}10]$ direction to form a tunnel. No anisotropic-isotropic transition took place in this system, instead, the unidirectional growth along GaAs $[\bar{1}10]$ direction that occurred was distinct from the results we reported here.

Comment 9: Again, this is a well-written paper which should be understood and appreciated by a wide readership.

Response: Thank you very much for this positive comment.

Reviewers' Comments:

Reviewer #1:

None

Reviewer #2:

Remarks to the Author:

The manuscript has been revised considering the comments. I recommend the publication in Nature Communications.

Reviewer #3:

Remarks to the Author:

As mentioned in my previous comments, I'm mainly concerned about the novelty of the research presented in this manuscript. The authors have now included ELO of III-V materials in their discussion and reply (which I gave as a single example), and emphasise that GE ELO is unique because of the non-polarity of the crystal. Much more work has, however, been done also on group IV ELO. See for instance:

RESEARCH ARTICLE | JANUARY 04 2008

Facet formation and lateral overgrowth of selective Ge epitaxy on μ -patterned Si(001) substrates

Ji-Soo Park; Jie Bai; Michael Curtin; Mark Carroll; Anthony Lochtefeld

Crossmark: Check for Updates

Author & Article Information

J. Vac. Sci. Technol. B 26, 117–121 (2008)

<https://doi.org/10.1116/1.2825165>

In this paper ELO of Ge has been discussed, including the formation of voids, for an isotropic crystal. Again, this is just a single example. I think it would be appropriate to summarize the (complete) state-of-the-art in the manuscript, and make the novelty of the current work clear.

Reviewer #4:

Remarks to the Author:

My concerns about this manuscript have been adequately addressed, and I approve of its publication.

Response to reviewers' comments

Reviewer #3 (2nd round):

As mentioned in my previous comments, I'm mainly concerned about the novelty of the research presented in this manuscript. The authors have now included ELO of III-V materials in their discussion and reply (which I gave as a single example), and emphasized that GE ELO is unique because of the non-polarity of the crystal. Much more work has, however, been done also on group IV ELO. See for instance: *J. Vac. Sci. Technol. B* **26**, 117–121 (2008)

In this paper ELO of Ge has been discussed, including the formation of voids, for an isotropic crystal. Again, this is just a single example. I think it would be appropriate to summarize the (complete) state-of-the-art in the manuscript, and make the novelty of the current work clear.

Response: We appreciate the importance of distinguishing the research in the manuscript from the current state of the art in ELO. For III-V, refs. 17 and 40-42 in the previous version of the manuscript describe overgrowth of lithographically patterned voids and ELO over mask areas. For group IV, to include a complete summary of ELO, we cross-checked results in the citation suggested (Park et al., *JVST B* **26**, 117 2008) and searched literature for others: Park et al., *Electrochem. Solid-State Lett.* **12** H142 (2009); Zaumseil et al. *J. Appl. Phys.* **106**, 093524 (2009); Leonhardt et al., *J. Crystal Growth* **335**, 62 (2011); Nam et al., *J. Crystal Growth* **416**, 21 (2015); Yako et al., *IEEE JSTQE* **24**, 8201007 (2018); and reviewed papers citing these publications. We now discuss this literature to clarify the novelty of the current work on **pages 2-4 and 17** of our manuscript with citations 7-12.

We agree with the referee that ELO has received detailed and careful study in both group III-V and group IV. In group IV ELO, the novelty of our manuscript arises because published works primarily focus on use of ELO to improve epilayer quality, for example reducing threading dislocation through aspect ratio trapping. This involves epilayer growth through thick oxide masks patterned into closely spaced trenches that have depth greater than width (and width comparable to our mask diameters). We identify only four papers that mention or discuss cavities in group IV ELO: *JVST* (2008) *JCG* (2011); *JCG* (2015) and *IEEE JSTQE* (2018). Park *et al.* (*JVST B* 2008), employing trenches along Si[110] and Si_{0.9}Ge_{0.1} as a marker layer, showed growth front evolution within the trench and across the top of the oxide. Voids were noted along the top of the oxide where the growth fronts met and their presence was attributed to Ge/oxide interface energy minimization. Leonhardt *et al.* (*JCG* 2011) showed that coalescence leads to twin defects, and mentioned that cavities form over oxide strips along Si[100] but not [110]. Nam *et al.* (*JCG* 2015) emphasized that cavities were unwanted, and discussed temperature and orientation conditions for cavity formation. Yako *et al.* (*IEEE JSTQE* 2018) observed cavities and reported improved epi-Ge layer quality with dislocations terminating at the cavity wall. These papers did not consider the compact dielectric masks whose use allowed us to observe and quantify the anisotropic-isotropic transition and derive behavior that is independent of the mask shape. Our manuscript demonstrates arrays of open pockets or cavities with potential applications and establishes an understanding of the observed phenomena through modelling. These comprise unique aspects of the paper that differentiate it from the existing literature and provide novelty for the current manuscript.

Reviewers' Comments:

Reviewer #3:

Remarks to the Author:

my comments have been addressed adequately